# Single-nucleus RNA-seq identifies transcriptional heterogeneity in multinucleated skeletal myofibers

Michael J. Petrany[1], Casey O. Swoboda[1], Chengyi Sun [1], Kashish Chetal[2], Xiaoting Chen [3], Matthew T. Weirauch [2,3,4], Nathan Salomonis [2,4] & Douglas P. Millay [1,4✉]

While the majority of cells contain a single nucleus, cell types such as trophoblasts, osteo-clasts, and skeletal myofibers require multinucleation. One advantage of multinucleation can be the assignment of distinct functions to different nuclei, but comprehensive interrogation of transcriptional heterogeneity within multinucleated tissues has been challenging due to the presence of a shared cytoplasm. Here, we utilized single-nucleus RNA-sequencing (snRNA-seq) to determine the extent of transcriptional diversity within multinucleated skeletal myofibers. Nuclei from mouse skeletal muscle were profiled across the lifespan, which revealed the presence of distinct myonuclear populations emerging in postnatal development as well as aging muscle. Our datasets also provided a platform for discovery of genes associated with rare specialized regions of the muscle cell, including markers of the myo-tendinous junction and functionally validated factors expressed at the neuromuscular junc-tion. These findings reveal that myonuclei within syncytial muscle fibers possess distinct transcriptional profiles that regulate muscle biology.

[1] Division of Molecular Cardiovascular Biology, Cincinnati Children's Hospital Medical Center, Cincinnati, OH, USA. [2] Division of Biomedical Informatics, Cincinnati Children's Hospital Medical Center, Cincinnati, OH, USA. [3] Center for Autoimmune Genomics and Etiology, Cincinnati Children's Hospital Medical Center, Cincinnati, OH, USA. [4] Department of Pediatrics, University of Cincinnati College of Medicine, Cincinnati, OH, USA. ✉email: douglas.millay@cchmc.org

Skeletal muscle forms through the fusion of mononucleated muscle progenitor cells during development[1,2]. Mature, multinucleated myofibers are composed of different contractile and metabolic machinery that result in independent muscle fiber types (IIb, IIx, IIa, and I)[3]. In addition to possessing fiber type diversity, skeletal muscle also contains a stem cell population (satellite cells) that can be activated upon a stimulus and ultimately fuse with myofibers or each other, thereby allowing regeneration[4]. The requirement of multinucleation for skeletal muscle function lacks experimental validation, but explanations include that each nucleus is capable of controlling a finite volume of cytoplasm and that regions of the myofiber perform specialized functions necessitating localized transcripts.

Compelling recent studies have enhanced our understanding of the transcriptional diversity of skeletal muscle. Single-cell RNA-sequencing and mass cytometry studies have outlined the major mononuclear cell types present in skeletal muscle, ranging from 10 to 15 cell types depending on cluster assignments and the granularity of subtyping[5–7]. The major cell types always include the following broad categories: fibroadipogenic progenitors (FAPs), tenocytes, endothelial cells, smooth muscle cells, immune cells (B cells, T cells, macrophages, neutrophils), neural/glial cells, and satellite cells. Notably, mature myofibers are a small minority of captures in single-cell studies despite their dominance within the tissue, owing to the fact that multinucleated cells are not easily isolated in single-cell dissociation approaches[5–7].

Myofibers are known to exhibit functional specialization at postsynaptic endplates (neuromuscular junction: NMJ), where a small cluster of myonuclei are responsible for formation and maintenance of the synaptic apparatus[8–11]. Postsynaptic myonuclei possess a unique transcriptional program that is highly specific and dissimilar to that of other myonuclei in the fiber, expressing acetylcholine receptor (AChR) subunit genes as well as many other enriched factors such as *Musk*, *Lrp4*, *Colq*, and *Etv5*[8]. Many of these factors have established roles in NMJ development, although more information regarding the full network of genes responsible for postsynaptic function could be revealed by more targeted and deeper sequencing approaches. Muscle-tendon connection sites (myotendinous junction: MTJ) are also known to exhibit structural specialization[12]. A number of structural proteins are enriched at the MTJ, including collagen XXII, α-7 integrin, and α-2 laminin, and mutations in their associated genes have been associated with myopathies in preclinical models or patients[13–15]. Nonetheless, the cell-specific regulation of the MTJ apparatus is poorly understood, and a transcriptionally unique population of myonuclei localized at the MTJ has not been defined.

Beyond the NMJ and MTJ, whether additional regions of compartmentalization are present in skeletal muscle is also not understood, likely because of technical limitations related to assessing transcription in syncytial cells. Stochastic transcriptional pulsing of particular genes has been reported in myofibers, but it is unknown whether this phenomenon reflects true divergence of myonuclear identities[16]. Transcriptional diversity may also be present in physiological contexts that elicit myonuclear accretion from satellite cells, such as in exercise[17]. Thus, the full extent of myonuclear heterogeneity achieved within syncytial myofibers remains unknown, but understanding this phenomenon will elucidate mechanisms that control muscle development and function. Here we use single-nucleus RNA-sequencing to investigate the transcriptional diversity of myonuclei in mouse skeletal muscle throughout the lifespan, revealing rare and previously unknown myonuclear populations as well as establishing an atlas of gene expression across postnatal development and aging.

## Results

We sought to gain insight into the range of myonuclear heterogeneity in mammals, through profiling of nuclei from mouse skeletal muscle using single-nucleus RNA-sequencing (snRNA-seq). Nuclei from 5-month mouse tibialis anterior muscle from wild-type mice were purified and the 10X Chromium system was used to build libraries for sequencing (Fig. 1a). We profiled 8331 nuclear transcriptomes, from which unbiased clustering revealed all major cell types expected in skeletal muscle, including myonuclei, satellite cells, fibroadipogenic progenitors (FAPs), endothelial cells, smooth muscle cells, tenocytes, and immune cells (Fig. 1b). Non-myonuclear clusters were identified through expression of canonical marker genes including *Pecam1* (endothelial cells), *Myh11* (smooth muscle), *Dcn* (FAPs), *Mkx* (tenocytes), and *Ptprc* (immune cells) (Supplementary Figs. 1, 2, Supplementary Data 1). We identified several clusters of myonuclei belonging to multinucleated muscle fibers, characterized by expression of numerous canonical myonuclear transcripts such as *Ttn*, *Tnnt3*, *Mybpc1*, and *Mybpc2* (Fig. 1c, Supplementary Fig. 2), and constituting a population essentially excluded by conventional single-cell analyses of skeletal muscle[5–7]. Myonuclei could be further distinguished by muscle fiber type, determined by expression of distinct members of the myosin heavy chain gene family. The tibialis anterior is a fast-twitch muscle known to express predominantly the fast myosin heavy chain isoforms *Myh4* (Type IIb fibers) and *Myh1* (Type IIx fibers)[18]. Consistent with this, the four largest clusters of myonuclei were grouped by expression of these two markers (Fig. 1c). A small number of *Myh2*[+] myonuclei (Type IIa) were also clustered upon higher-dimensionality analysis (Supplementary Fig. 3, Supplementary Data 1)[19], although Type IIa myofibers are known to comprise only a small minority within the mouse tibialis anterior[18].

Myonuclei that clustered outside of fiber type included those of the neuromuscular and myotendinous junctions. 0.8% of myonuclei were identified as belonging to the neuromuscular junction (NMJ), where postsynaptic transcriptional specialization is known to occur. NMJ nuclei were positive for known canonical markers such as *Chrne*, *Etv5*, *Colq*, and *Musk* (Fig. 1c and Supplementary Fig. 4a), but also expressed numerous genes not previously associated with the NMJ, with a total of 345 significantly upregulated genes (Supplementary Data 2). Myotendinous junction (MTJ) nuclei comprised 3.6% of myonuclei and were notable for expression of *Col22a1* and *Ankrd1* (Fig. 1c and Supplementary Fig. 4b), whose expression has been reported to concentrate at the ends of muscle fibers[13,20]. Despite data showing that myonuclei may cluster near the MTJ[21,22], robust evidence of a unique nuclear population with a specific transcriptional program has not been shown. Here, we identified 291 uniquely upregulated genes, and genes previously not associated with the MTJ included *Slc24a2* and *Adamts20* (Supplementary Fig. 4b, Supplementary Data 2). The source of *Col22a1* at the ends of myofibers is not clear, but recent evidence indicates contribution from a tenocyte population[23]. Our data show *Col22a1* and myosins are co-expressed, as well as *Col22a1* expression in tenocytes (Fig. 1c), suggesting both myonuclei and tenocytes contribute this extracellular matrix protein. We validated expression of *Col22a1* and *Adamts20* at the MTJ in tibialis anterior muscle using single-molecule FISH (smRNA-FISH), confirming the localization of this myonuclear population (Fig. 1d). These data indicate strong spatial transcriptional heterogeneity of NMJ and MTJ myonuclei, suggesting that location within the syncytia and associated proximal signaling are major determinants of transcriptional profiles.

In addition to transcriptional specialization at the NMJ and MTJ, we asked whether any further heterogeneity within fiber type

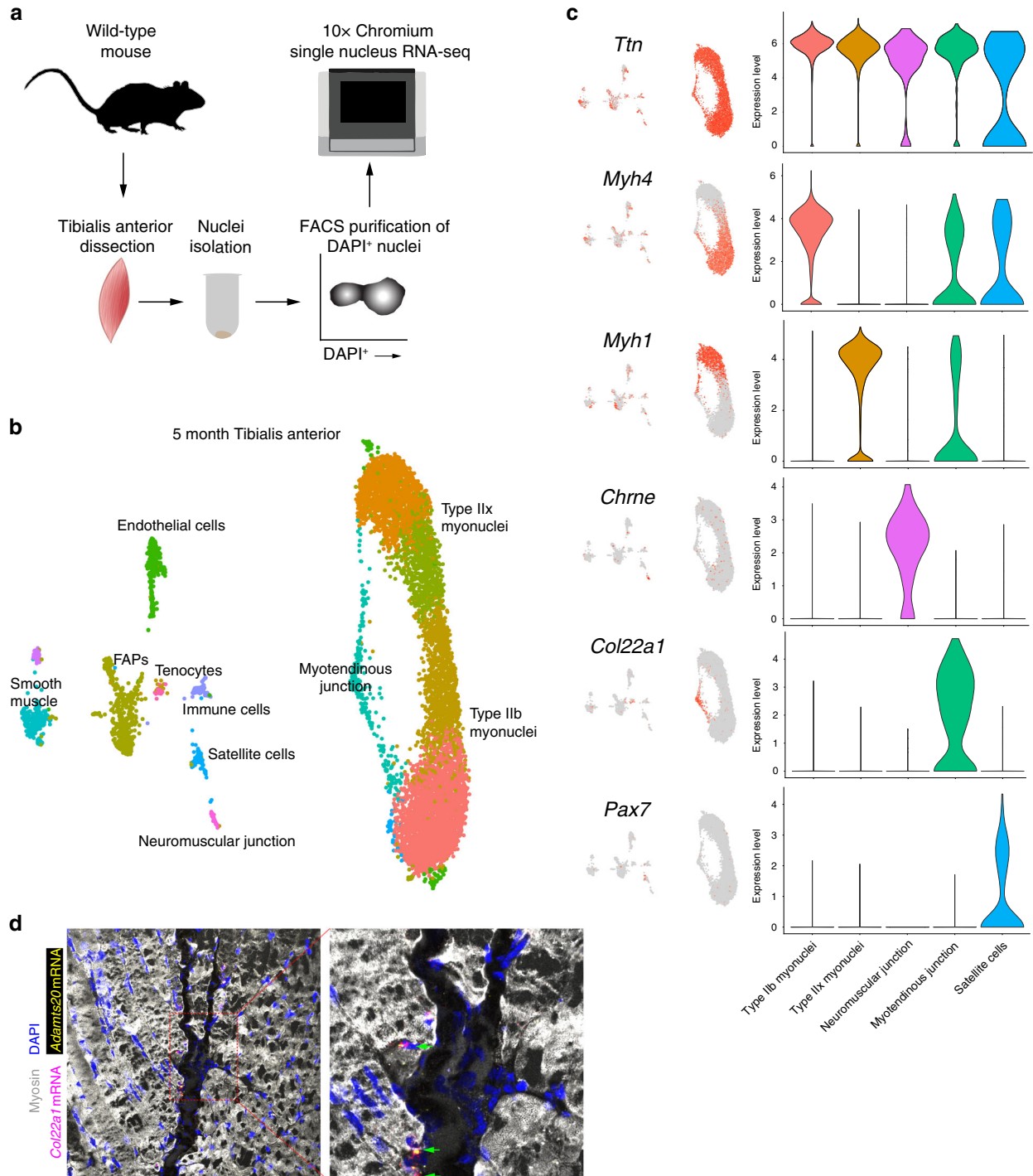

**Fig. 1 snRNA-seq of mouse tibialis anterior muscle at 5 months of age. a** Schematic for nuclei purification and sequencing from mouse skeletal muscle. **b** Unbiased clustering of snRNA-seq data represented on a UMAP. **c** UMAP and violin plots showing gene expression for myonuclear populations including *Ttn* (all myonuclei), *Myh4* (Type IIb myonuclei), *Myh1* (Type IIx myonuclei), *Chrne* (neuromuscular junction), *Col22a1* (myotendinous junction), and *Pax7* (satellite cells). Intermediate myonuclear clusters were combined with their respective fiber type myonuclei for generation of violin plots. The *y*-axis shows expression level as probability distribution across clusters. **d** Representative image of smRNA-FISH for *Col22a1* and *Adamts20* on 5-month longitudinal tibialis anterior sections, showing localization of transcripts in myonuclei at the myotendinous junction ($n = 5$). Green arrows show *Col22a1*[+] *Adamts20*[+] nuclei. Scale bars: 50 μm (left panel), 10 μm (right panel).

specific myonuclei was discernible through subclustering of exclusively *Myh4*[+] (Type IIb) or *Myh1*[+] (Type IIx) myonuclei. Gene Ontology enrichment analysis confirmed that these nuclear populations are enriched in cellular component terms associated with the sarcomere and muscle contraction, while terms associated with mitochondria were selectively present in *Myh1*[+] nuclei, which

are known to have an increased oxidative metabolic profile (Supplementary Fig. 5a). Subclustering revealed multiple nuclear compartments with divergent transcriptional states (Supplementary Fig. 5b, c). While these clusters did not possess many unique markers, they did exhibit a spectrum of gene expression potentially indicating modest transcriptional heterogeneity in homeostatic

muscle. We then investigated the possibility of stronger myonuclear subtypes with distinct functions utilizing different types of skeletal muscle and at varying ages.

To test if myonuclear heterogeneity is dependent upon a particular muscle type, we profiled 5389 nuclei from the slow-twitch soleus muscle. Here, the populations detected were consistent with those in the tibialis anterior but also included Type I myonuclei (*Myh7*$^+$), which is characteristic of the soleus muscle (Supplementary Fig. 6, Supplementary Fig. 7a, b)[18]. Integration of the tibialis anterior and soleus data resulted in a comprehensive atlas of fiber type expression profiles and showed that Type I myonuclei are the most divergent likely owing to unique activities of this type of muscle (Supplementary Fig. 7c, d).

While our analysis of 5-month muscle suggests the possibility of previously unknown myonuclear diversity, outside of the NMJ and MTJ, it was not clear if the level of diversity indicated true functional compartmentalization. We asked whether more prominent myonuclear heterogeneity might be evident during postnatal development. Robust accrual of myonuclei during postnatal development typically ceases by 21 days in healthy mice, by which time the myogenic differentiation program has been switched off and myofibers possess nearly their full complement of myonuclei[24]. However, muscle growth and maturation continue beyond this point until mice reach adulthood, and how this process of muscle growth impacts transcription in accrued myonuclei is unclear. We profiled 11,552 nuclear transcriptomes from postnatal (P) day 21 tibialis anterior muscle to assess transcriptional dynamics of maturing myonuclei. P21 myonuclei clustered in a pattern distinct from that of adult mice, and clusters were not solely reducible to fiber type identity (Fig. 2a, Supplementary Fig. 8). In addition to Type IIb, Type IIx, NMJ, and MTJ myonuclei, two additional myonuclear populations were identified. One of the developmental clusters of myonuclei were characterized by expression of *Enah* and the other by expression of *Nos1* (Supplementary Fig. 9a). Further clustering of only myonuclei showed more specific subpopulations that were clearly distinguished from the Type IIb and Type IIx myonuclei (Fig. 2b). Differential gene-expression analysis revealed two clusters of *Ttn*$^+$ myonuclei that less strongly expressed markers of end-state myonuclear differentiation (*Myh1, Myh4, Ckm*) (Fig. 2b, Supplementary Fig. 9b). These *Myh4*$^-$/*Myh1*$^-$ clusters showed a skeletal muscle-specific gene signature (Supplementary Fig. 10), but did not highly express myogenic markers (*Myod1, Myog*, and *Mymk*) suggesting they represent nuclei within myofibers (Fig. 2c). Three additional populations were identified: two displayed scattered expression of *Myh4* or *Myh1* (Fig. 2c and Supplementary Fig. 9b), which we assigned fiber type-like designations; and a small unknown *Meg3*$^+$ population. The *Myh4*$^-$/*Myh1*$^-$ clusters displayed unique transcriptional programs and were enriched for factors associated with early myofibrillogenesis, including *Nrap, Fhod3, Enah*, and *Flnc*, as well as the non-muscle myosins *Myh9* and *Myh10*, which have been postulated as components of "pre-myofibrils" laid down as scaffolds before mature muscle-specific myosins are assembled (Fig. 2d)[25–30]. We therefore referred to these populations as sarcomere assembly states. Multiple transcription factors were enriched in sarcomere assembly myonuclei, including both known drivers of skeletal muscle differentiation (*Ifrd1, Nfat5, Mef2a*)[31–33] and numerous TFs with no previous associations in muscle (*Ell, Creb5, Zfp697*) (Supplementary Fig. 9c). We performed TF binding site enrichment analysis on the upregulated genes within each cluster, and discovered that Atf3 motifs were significantly enriched near transcriptional start sites of marker genes of one of the sarcomere assembly states (Supplementary Fig. 11a), and that *Atf3* transcripts were highly increased in that same population in our snRNA-seq data (Supplementary

Fig. 11b). We confirmed that *Atf3* mRNA is enriched in nuclei with high levels of *Flnc* transcription in P21 tibialis anterior muscle by smRNA-FISH (Supplementary Fig. 11c). Altogether, these results provide evidence for a coordinated transcriptional regulatory program and suggest a previously unknown role for Atf3 in myonuclear function during development. Thus, these data show temporal myonuclear heterogeneity within skeletal muscle at P21 and indicate that an expanded pool of myonuclei acquire a distinctive, specialized signature by which coordinated transcriptional activity supports increased myofibrillogenesis and sarcomere assembly.

One interpretation of P21 sarcomere assembly myonuclei is that they are the most recently fused nuclei and have not yet established their fiber type. In this scenario, these sarcomere assembly myonuclei could be in a more immature state and suggest that they progress through a postfusion maturation program to establish their adult identity. Another possibility is that these myonuclear states are entrained after the majority of myonuclear accretion has occurred (around P21). To distinguish these possibilities, we generated 8611 nuclear transcriptomes from the tibialis anterior at P10 when fusion is ongoing and recently fused nuclei would be present. Here we were unable to detect robust independent clustering of *Enah*$^+$ myonuclear populations and instead detected myonuclear populations similar to adult muscle (Fig. 3a, Supplementary Fig. 12a). Interestingly, we did detect a cluster of cells at P10, which were close to satellite cells, but were not present at P21 or 5 months of age. We hypothesized that this population represented myocytes (activated satellite cells) since fusion is ongoing at P10. We subclustered the satellite cell and myocyte populations and detected the presence of differentiation and fusion genes including *Myog* and *Mymk* in the myocyte population, while *Pax7* was enriched in the satellite cell population (Supplementary Fig. 12b). The presence of myocytes demonstrated ongoing fusion at P10 and the absence of this population at P21 confirmed that the majority of fusion was terminated in these samples. The lack of sarcomere assembly populations at P10 indicates that they do not represent a maturational phase that myonuclei transition through proximal to fusion, but instead more nuclei adopt this state in response to a physiological stimulus during the postfusion growth of skeletal muscle. We validated the transient expansion of *Enah*$^+$ myonuclei at P21 by performing smRNA-FISH on tibialis anterior muscles from P10, P21, and 5-month-old mice (Fig. 3b). *Flnc* and *Enah* expression was present at all ages, but the percentage of *Flnc*$^+$/*Enah*$^+$ nuclei was significantly increased in P21 compared to P10 and 5 months (Fig. 3b). This result validated the P21 snRNA-seq data while also confirming our initial observation of low-level heterogeneity and *Enah*$^+$ myonuclei in adult muscle (Supplementary Fig. 5c).

In addition to functional diversity found in normal development, transcriptional heterogeneity can arise from pathological or compensatory processes associated with aging[34–36]. Skeletal muscle undergoes dramatic age-related changes with respect to functional deterioration and regenerative decline[37–40], and so we next asked if the aging process impacted myonuclear transcription. We profiled 18,087 nuclei from 24- and 30-month-old tibialis anterior muscles (Fig. 4a, b, Supplementary Figs. 13 and 14). In 30-month muscle only, we observed emerging myonuclear populations that were not present in 5 or 24 months, including clusters expressing *Nr4a3, Ampd3*, and *Enah*, respectively (Fig. 4b). We integrated myonuclei from 5, 24, and 30 months and found a consistent aging-related gene-expression signature including upregulation of *Nr4a3* and *Smad3* at both aged time-points (Fig. 4c, Supplementary Data 3), indicating that this may be a progressive feature of aging muscle. *Smad3* has been implicated during the muscle aging process[41,42], while *Nr4a3* has a role for

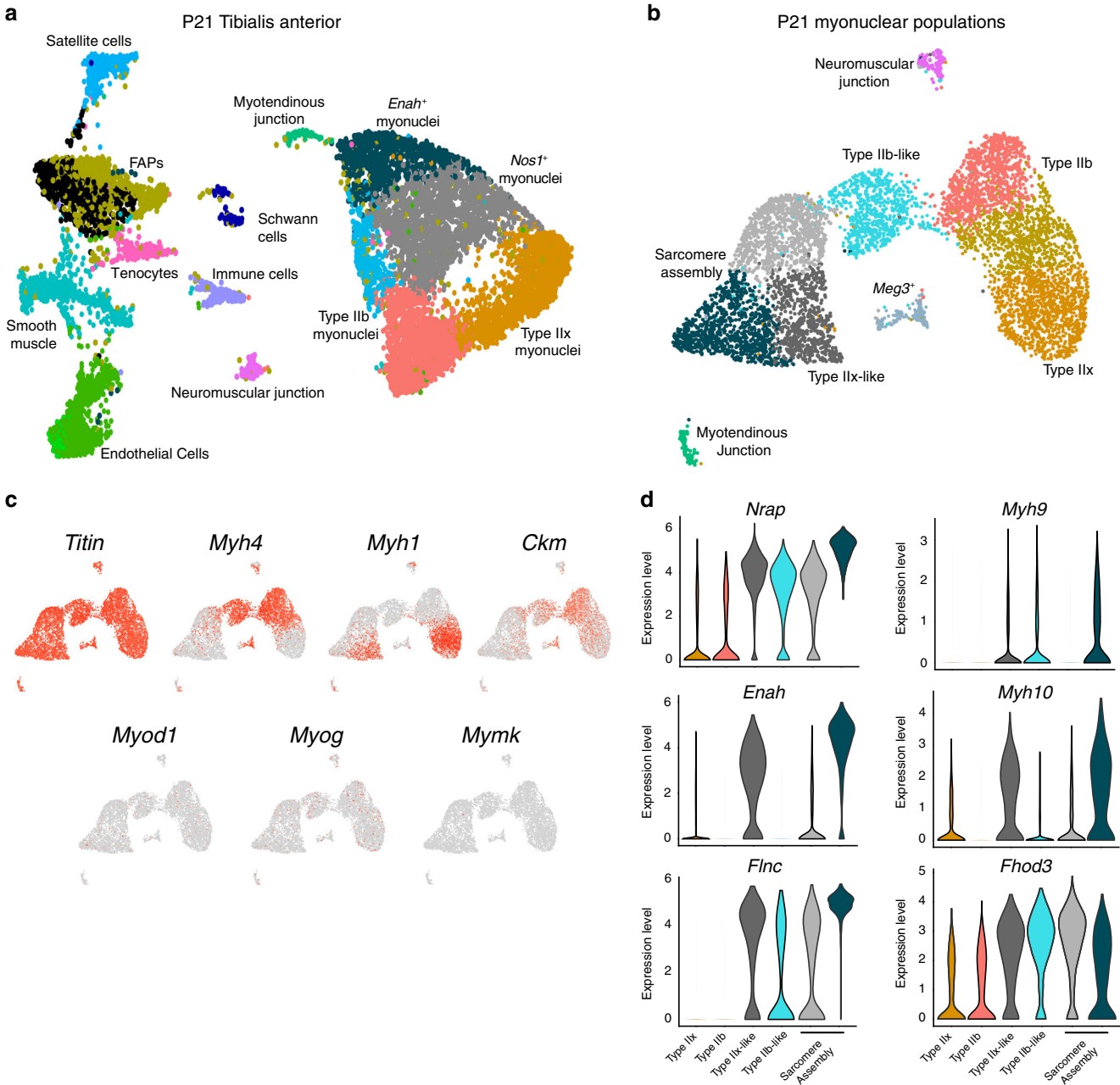

**Fig. 2 Temporal myonuclear heterogeneity revealed in developing muscle. a** Unbiased clustering of nuclei from postnatal (P) day 21 presented in a UMAP revealed all major populations in addition to unique myonuclei outside the canonical Type IIb and Type IIx myonuclei. These nuclei populations were marked by *Nos1* and *Enah,* respectively. **b** Subclustering of myonuclear populations from P21 muscle revealed sarcomere assembly myonuclear states. **c** Feature plots assessing expression of mature muscle markers (*Ttn, Myh1, Myh4,* and *Ckm*) and differentiation markers (*Myod1, Myog, Mymk*) from the myonuclear populations in **b**. **d** Violin plots showing the sarcomere assembly myonuclei are enriched for a transcriptional profile associated with early myofibrillogenesis (expression of *Nrap, Enah, Flnc, Myh9, Myh10, Fhod3*).

metabolic adaptations to exercise but has not been studied during aging[43,44]. smRNA-FISH for *Nr4a3* in the gastrocnemius muscle confirmed elevated expression in myonuclei in aged muscle, following a heterogeneous pattern (Fig. 4d). The *Ampd3*+ population were identified as myonuclei based on the presence of the myogenic regulatory factor *Myf6*, the pro-atrophy gene *Fbxo32*, and neuromuscular markers *Musk, Chrnb1,* and *Hdac4* (Supplementary Fig. 15). The cluster was also enriched for genes associated with the immune response, apoptosis, and proteasomal degradation (*Tnfrsf23, Traf3,* and *Psma5*) (Supplementary Fig. 15). Taken together, these data suggest that the *Ampd3*+ population could represent a denervated state that is dysfunctional (Supplementary

Fig. 15). Surprisingly, the *Enah*+ myonuclear cluster was found to be highly similar to the *Enah*+ sarcomere assembly state from P21 muscle, with numerous marker genes in common including *Atf3, Flnc,* and *Nrap* (Fig. 4e). The presence of an *Enah*+ cluster at 30 months presents further evidence for the appearance of this sarcomere assembly population throughout the lifespan, appearing most prominently in development and aging.

In order to more systematically investigate the presence of sarcomere assembly myonuclei over time, as well as to determine patterns of nuclear composition across the lifespan, we integrated our snRNA-seq data from the tibialis anterior at P10, P21, 5 months, 24 months, and 30 months (Fig. 5a). Notably, integration

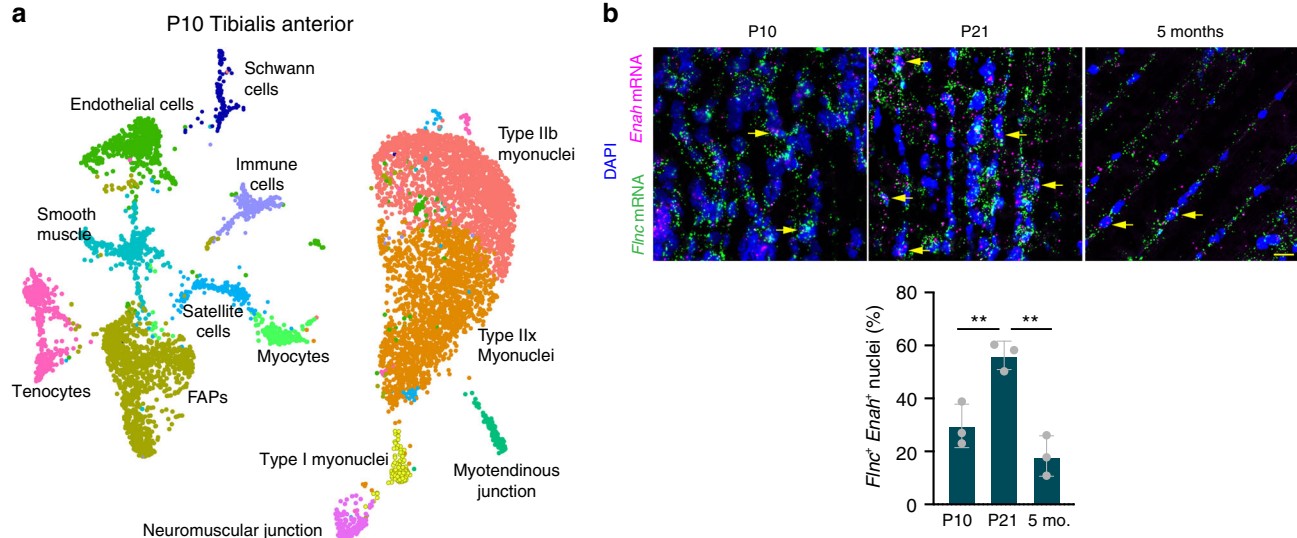

**Fig. 3 Myonuclear transcriptional states exhibit temporal specificity in development. a** UMAP representing snRNA-seq data from a postnatal (P) day 10 tibialis anterior shows the presence of an activated muscle progenitor population (myocytes) but the absence of defined sarcomere myonuclear states. **b** smRNA-FISH for *Flnc* and *Enah* shows an expansion of sarcomere assembly state myonuclei in P21 tibialis anterior compared to P10 and adult muscle. *Flnc*+ *Enah*+ nuclei were quantified at each timepoint (n = 3). Data are presented as mean ± standard deviation. A one-way ANOVA with Tukey post-hoc comparison was used to determine statistical significance, **$P < 0.01$. Scale bar: 20 μm. Source data are provided as a Source Data file.

confirmed the independent clustering of *Enah*+ and *Ampd3*+ populations (Fig. 5a). *Enah*+ myonuclei were present in all time-points but increased in P21 muscle (Fig. 5b), consistent with our smRNA-FISH findings and analysis of individual datasets. Intriguingly, we also identified a population of nuclei (*Ckm*+ *Col1a1*+) expressing a number of skeletal muscle-specific genes (e.g., *Tnni2, Tnnt3, Ckm*) as well as marker genes involved in extracellular matrix remodeling (e.g., *Col1a1, Col3a1, Col5a3, Col6a1*, and *Dcn*) (Fig. 5a, Supplementary Data 1). This population was over-represented in development, comprising 4.7% of nuclei at P10 compared to 0.9% of nuclei at 5 months (Fig. 5b). This cluster could represent an intermediate population that shares characteristics of myonuclei as well as cells of fibroblastic or tendon origin. We confirmed that our combined datasets are likely to represent a faithful trajectory of development and aging, as an independent approach for combined analysis without batch correction found overlapping cells in all clusters, indicative of temporally related cell states (Supplementary Fig. 16).

In addition to revealing the emergence of shared transcriptional responses at distinct stages of the life cycle, our data also establish an atlas of myonuclear transcriptomics and reveal transcripts associated with rare myonuclear populations such as the NMJ. Previous studies have elegantly uncovered unique gene expression in NMJ-associated nuclei, which allowed for identification of fundamental molecular components of postsynaptic function[45–47]. However, our data reveal numerous previously unreported genes associated with the NMJ, likely because our approach was able to resolve global transcription at the nuclear level. This represents a unique opportunity to further reveal mechanisms of postsynaptic development and function. To test the validity of genes not previously associated with the NMJ, we confirmed localization of 4 of the genes not previously associated with the NMJ (*Ufsp1, Lrfn5, Ano4*, and *Vav3*) by smRNA-FISH and co-labeling of the NMJ marker acetylcholine receptor (AchR) (Fig. 6a). Moreover, we found that multiple top NMJ genes identified here showed significant changes in mRNA levels following hindlimb denervation, an expression pattern associated with postsynaptic regulation (Fig. 6b)[48]. While the expression of the majority of those genes increased after denervation, similar to the canonical NMJ factor

*Musk*[49], *Ufsp1* and *D430041D05Rik* exhibited a decrease in expression, while *Pdzrn4* did not change (Fig. 6b). To assess the functional relevance of these factors for NMJ formation and maintenance, we used an in vitro culture system in which C2C12 myoblasts, when plated on laminin, form aneural NMJ-like clusters that are marked by AChR (Fig. 6c)[50–52]. We treated C2C12 cells with a control siRNA or siRNAs targeting *Musk, Ufsp1, Vav3, B4galnt3*, or *Gramd1b*, then assayed for AChR clustering. Reduced expression of the target genes was observed after treatment with the appropriate siRNA (Fig. 6d). *Musk* reduction results in a blockade of AChR clustering (Fig. 6e), showing fidelity of the system[50]. Obvious AChR clustering abnormalities were not observed for cultures where *Vav3* or *B4galnt3* was reduced, but *Ufsp1* and *Gramd1b* were identified as regulators of the NMJ (Fig. 6e). Loss of *Gramd1b* resulted in reduced AChR clusters, whereas reduction of *Ufsp1* elicited an increase (Fig. 6f). These data indicate that *Ufsp1* is a negative regulator of AChR clustering and NMJ formation, consistent with its downregulation following denervation. Reduction of *Ufsp1* or *Gramd1b* did not impact overall myogenesis or transcript levels of canonical NMJ genes (Fig. 6g), suggesting they specifically regulate the NMJ in a transcriptionally independent manner. These data highlight the utility of nuclear-level resolution for the discovery of factors that regulate the biology of skeletal muscle.

## Discussion
Altogether, our single-nucleus profiling of skeletal muscle reveals intricate transcriptional dynamics of myonuclear states across the lifespan of the mouse. To this end, an interactive data portal is publicly available at https://research.cchmc.org/myoatlas. An unexpected finding was the presence of a unique sarcomere assembly transcriptional signature in a subset of myonuclei, most prominent in developing muscle after postnatal fusion and myonuclear accrual was terminated. We speculate that expansion of the sarcomere assembly state is determined by stage-specific growth signals following the tapering of developmental fusion, or is reflective of the magnitude of growth occuring during this phase. Further experiments will be required to identify the key

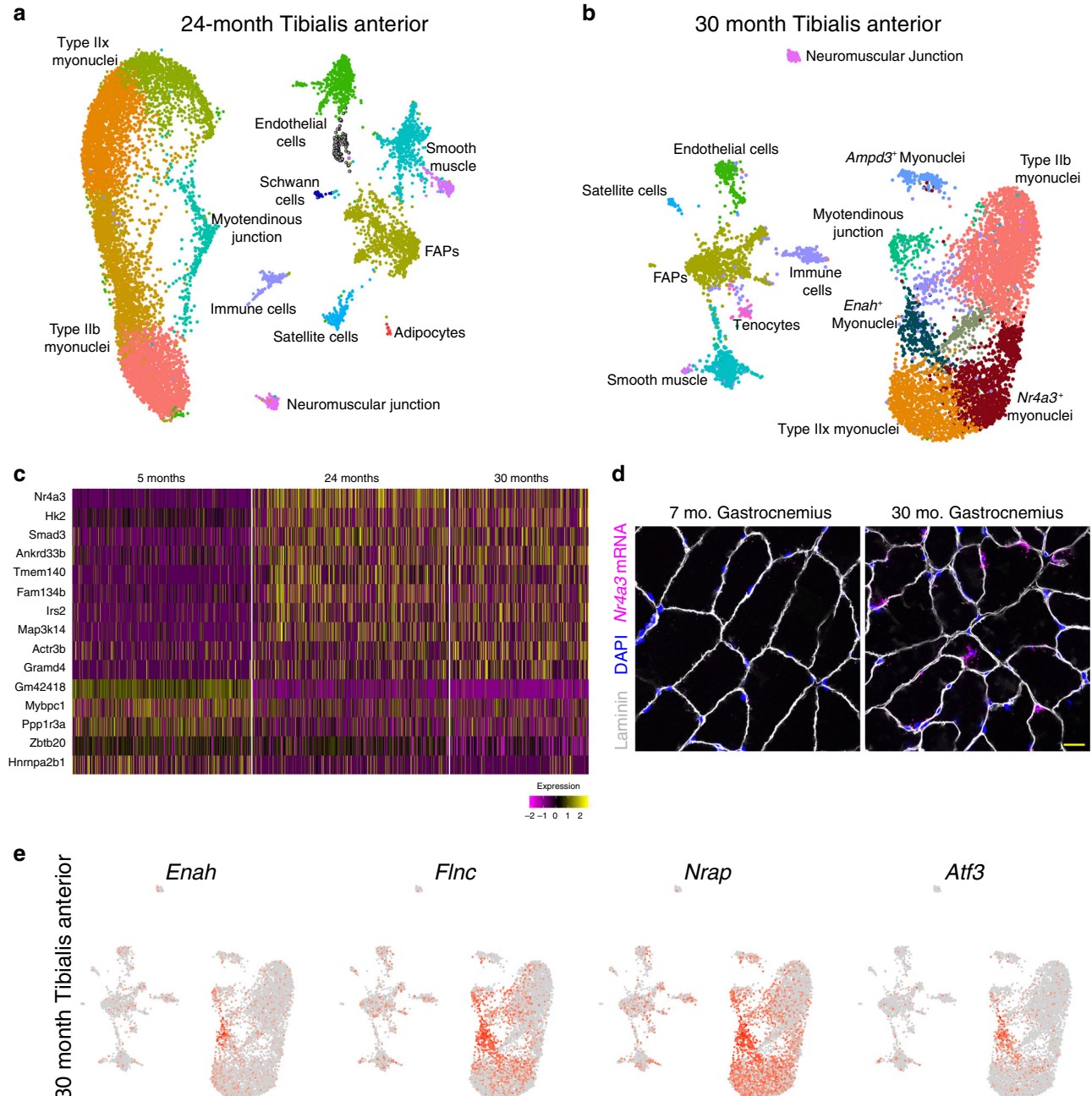

**Fig. 4 Uniform gene expression between myonuclei is disrupted in aged muscle. a** UMAP of snRNA-seq data from the tibialis anterior of a 24-month-old mouse displaying similar clusters as at 5 months of age. **b** UMAP of snRNA-seq data from 30-month-old muscle displaying the presence of unique clusters of myonuclei marked by *Ampd3, Enah*, and *Nr4a3*. **c** Heatmap of integrated myonuclei from 5, 24, and 30-month tibialis anterior muscle showing a consistent transcriptional signature of aging muscle. Columns are individual nuclei belonging to each to timepoint. **d** Representative image of smRNA-FISH for *Nr4a3* validated its upregulation in myonuclei in 30-month skeletal muscle (*n* = 3 for each group). Scale bar: 10 μm. **e** Feature plots of key marker genes of the *Enah*+ population within 30-month myonuclei.

regulatory factors that drive the sarcomere assembly myonuclear state and to elucidate its contribution to postnatal development and muscle maintenance.

Our analyses also provide a comprehensive resource for further study of rare but essential myonuclear populations. We identify numerous genes for the specialized NMJ and MTJ compartments and provide evidence that previously uncharacterized NMJ factors are functional regulators of postsynaptic function. Little is known regarding the regulatory mechanisms that drive NMJ-related transcription in postsynaptic myonuclei, while restricting this program in other nuclei within the syncytium. Interrogation of factors with demonstrated functional roles in NMJ formation, particularly those that may act in a negative regulatory capacity, will be critical to further understanding in this area. In addition, independent approaches now indicate the existence of a *Col22a1*+ population of myonuclei located within myofibers at the mammalian MTJ and provide robust transcriptomes for these nuclei[53]. These results should provide a powerful resource for addressing unresolved questions with respect to the cellular composition and transcriptional regulation of the MTJ, such as mechanisms of

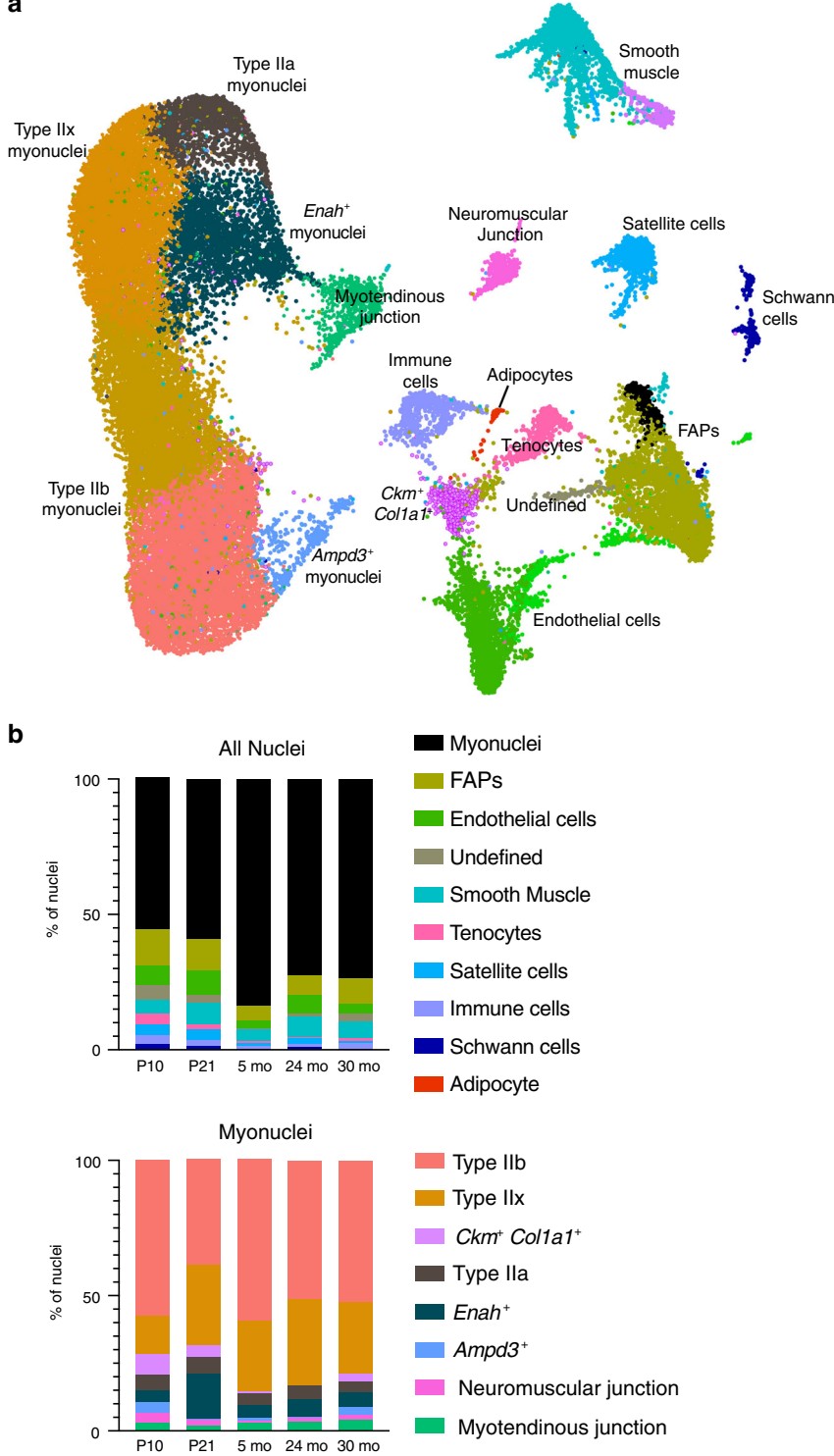

**Fig. 5 Integration of snRNA-seq across the lifespan reveals myonuclear population dynamics. a** Integrated UMAP of snRNA-seq data from postnatal day (P) 10, P21, 5-month, 24-month, and 30-month tibialis anterior muscle. **b** Proportional composition of nuclear types across the lifespan. Major categories (FAPs, endothelial cells, smooth muscle) include closely-related subpopulations. Myonuclei are combined (top graph) and subsetted (bottom graph) to display myonuclear composition dynamics.

crosstalk between myofibers and tendon cells[12] and the implication of MTJ-related factors in inherited myopathies[13–15].

Additionally, integration of tibialis anterior snRNA-seq data from all timepoints revealed a population of nuclei with muscle-specific markers but also a signature enriched for extracellular matrix-related genes such as *Col1a1, Col3a1, Col5a3, Col6a1,* and

*Dcn*. This population bears some similarity to the MTJ-B population reported in the accompanying manuscript[53], a second MTJ population in addition to the *Col22a1*[+] myonuclei identified in both studies. It is possible that the inclusion of transcriptionally similar non-muscle mononuclear cells in our data, for example FAPs and tenocytes, precludes the emergence of the second MTJ

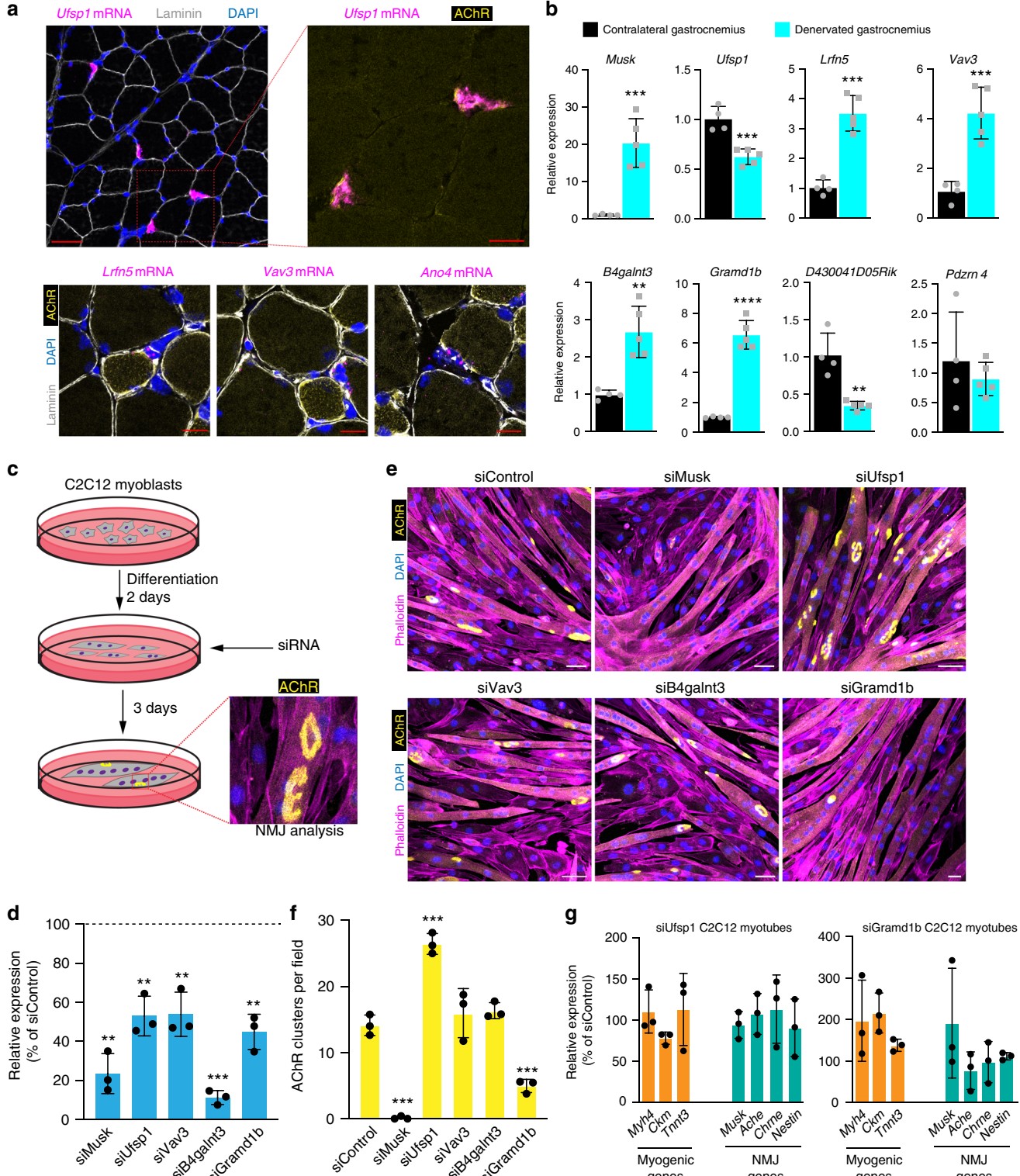

**Fig. 6 Discovery and functional characterization of previously unknown NMJ marker genes. a** Representative images from smRNA-FISH for *Ufsp1*, *Lrfn5*, *Vav3*, and *Ano4* on skeletal muscle sections showing co-localization with the canonical NMJ protein, acetylcholine receptor (AChR) (*n* = 3). AChR was visualized through α-bungarotoxin labeling. **b** Quantitative real-time PCR (qPCR) analysis for the indicated genes not previously associated with the NMJ from normal (*n* = 4) and denervated (*n* = 5) muscle. **c** Schematic for a siRNA screen in C2C12 myoblasts designed to test the function of candidate NMJ genes. siRNA was transfected two days after differentiation and three days later α-bungarotoxin was used to analyze AChR clustering as a surrogate for NMJ formation. **d** qPCR analysis for the genes targeted with siRNA. A scrambled siRNA was used as a control. **e** Representative images of C2C12 myotube cultures after treatment with various siRNAs and staining with α-bungarotoxin. Cells were also stained with phalloidin and DAPI. **f** Quantification of AChR clusters per field of view from **e**. **g** qPCR analysis for genes associated with myogenesis and NMJ formation. Scale bars: **a** 50 μm (top left panel), 10 μm (top right and bottom panels), **e** 50 μm. Data in **d**, **f**, and **g** are from three independent experiments. All data are represented as mean ± standard deviation. An unpaired two-sided *t*-test was used to determine statistical significance, **P < 0.01, ***P < 0.001, ****P < 0.0001. Source data are provided as a Source Data file.

population in our individual datasets, which instead appears only upon integration of all timepoints. Further experimentation would be necessary to confirm the correlation of these populations.

Our snRNA-seq successfully captured known nuclear diversity while also uncovering transcriptional states across all ages interrogated. While we analyzed all nuclei from muscle, approaches to enrich specifically for myonuclei[53] or to improve transcript detection levels may reveal further insights into myonuclear heterogeneity. One caveat of our work is the need for functional validation of previously unidentified myonuclear populations, which will be the focus of future investigations. Nonetheless, snRNA-seq can be leveraged towards greater understanding of myonuclear dynamics in pathological states such as acute or chronic injury or in human myopathies. Altogether, our snRNA-seq data have revealed rare myonuclear populations and provided a powerful resource for understanding of myonuclear transcriptional heterogeneity across the lifespan. Further studies will help to clarify the mechanisms that lead to assignment of certain nuclei to specialized compartments within a shared myofiber cytoplasm.

## Methods

**Mice and associated experimental procedures**. All of the mice used in this study were C57BL/6 wild-type males. Mice were housed at 22.2 °C with humidity of 30–70% and a 14 h light/10 h dark cycle. Mice were fed ad libitum and all snRNA-seq experiments were performed in the morning. For isolation of nuclei for snRNA-seq, tibialis anterior muscles were taken from a single mouse (5-month, 24-month, 30-month) or pooled from 2 mice (P21), or 4 mice (P10) to collect sufficient nuclei for sequencing. Soleus muscles were pooled from 4 mice. Unilateral hindlimb denervation was performed on C57BL/6 wild-type male mice by cutting the left sciatic nerve in the mid-thigh region after the mouse was anesthetized with isoflurane. The gastrocnemius muscle was harvested 3 days after denervation and the contralateral muscles were used as control. For smRNA-FISH experiments, muscles were dissected, embedded in 10% tragacanth/PBS (Sigma) for cross-sections or OCT compound (Fisher) for longitudinal sections, and frozen in 2-methylbutane cooled in liquid nitrogen. 10-μm sections were prepared for all experiments. For RNA isolation, muscles were flash-frozen in liquid nitrogen and stored at −80 °C until use. All mouse procedures were approved by Cincinnati Children's Hospital Medical Center's Institutional Animal Care and Use Committee (IACUC2017-0053).

**Purification of nuclei from mouse skeletal muscle**. Tibialis anterior or soleus muscles were isolated from mice immediately following euthanasia, minced, and placed in homogenization buffer (0.25 M sucrose and 1% BSA in $Mg^{2+}$-free, $Ca^{2+}$-free, RNase-free PBS). The minced tissue was further homogenized using an Ultra-Turrax T25. The homogenate was then incubated for 5 min with addition of Triton-X100 (2.5% in RNase-free PBS, added at a 1:6 ratio). Samples were then filtered through a 100-μm strainer, centrifuged ($3000 \times g$ for 10 min at 4 °C) to form a crude pellet, and resuspended in sorting buffer (2% BSA/RNase-free PBS) and filtered again through a 40 μm strainer. Hoechst dye was added to label nuclei as well as 0.2 U/μl Protector RNase inhibitor (Roche). Labeled nuclei were purified via FACS (BD Aria, 70 μm nozzle) and collected in sorting buffer containing RNase inhibitor. The resulting nuclear suspension was pelleted with a light centrifugation step ($250 \times g$ for 5 min at 4 °C) and the supernatant drawn off to concentrate the nuclei. Aliquots were taken for visualization under a fluorescent microscope to assess nuclear appearance and integrity.

**Construction of libraries and generation of cDNA on the 10X Genomics platform**. Nuclei were counted using a hemocytometer and the concentration adjusted if needed to meet the optimal range for loading on the 10X Chromium chip. The nuclei were then loaded into the 10X Chromium system using the Single Cell 3′ Reagent Kit v3 according to the manufacturer's protocol. We aimed to load ~13,000 nuclei for each run. Following library construction, libraries were sequenced on the Illumina NovaSeq 6000 system.

**Single-nucleus data analysis**. Raw sequencing data of all samples were processed using the cellRanger workflow (version 3.1.0), using a combined intron-exon reference produced as described using the vendor-provided "Generating a Cell Ranger compatible "pre-mRNA" Reference Package" guidelines (https://support.10xgenomics.com/single-cell-gene-expression/software/pipelines/latest/advanced/references). In brief, the "pre-mRNA" reference was derived using the default exon-level GTF file provided by 10x Genomics (http://cf.10xgenomics.com/supp/cell-exp/refdata-cellranger-mm10-3.0.0.tar.gz). Using the below awk command, this exon-level GTF file into "pre-MRNA" GTF containing intron transcript

definitions ($ awk 'BEGIN{FS = "\t"; OFS = "\t"} $3 = = "transcript"{$3 = "exon"; print}' genes.gtf > genes.premrna.gtf). Next, the below mkref command was run to produce the final "pre-MRNA" GTF and genome fasta file ($ cellranger mkref–genome=Mmpre–fasta=genome.fa–genes=genes.premrna.gtf). For each dataset, we corrected for ambient background RNA by filtering with the R package SoupX[54]. We used the inferNonExpressedGenes() function to determine which genes had the highest probability of being ambient mRNA, and the strainCells() function in order to transform count matrices. Genes expressed per nucleus was between 1446 and 2445, depending on the dataset. Quality control metrics for each dataset, including genes expressed per nucleus, are summarized in Supplementary Data 4.

Further data analysis was carried out using the R (version 3.6.1) package Seurat (version 3.1.0)[55]. For individual datasets, outlier nuclei were excluded based on unique feature counts. Nuclei with less than 200 expressed features were excluded, as well as nuclei with greater than 3200 expressed features (5-month dataset) or 4000 expressed features (all other datasets). No features that were expressed in 3 or fewer cells were included. Data from filtered nuclei were then normalized logarithmically using the NormalizeData() function. Using the FindVariableFeatures() function, 2000 features with high variable expression across the nuclei were identified and used in a subsequent Principal Component Analysis (PCA) using the RunPCA() function. For clustering and Uniform Manifold Approximation and Projection (UMAP) visualization, the number of components used was unique to each dataset, based on the strength of PCs visualized by elbow plot and PC heatmaps after PCA was performed. The UMAPs and clusters were generated using the FindNeighbors(), FindClusters(), and RunUMAP() functions (R implementation). Of note, the default resolution in this pipeline was 0.5. Violin and Feature plots were generated using the VlnPlot() and FeaturePlot() from the Seurat package, respectively. Heatmaps were generated using the DoHeatmap() function, displaying the top marker genes by highest average log fold change for each cluster. For subclustering, the function subset() was used to select populations of interest from the data and create a new Seurat object with just those identities, followed by standard Seurat preprocessing. In order to analyze the 5-month dataset with higher dimensionality, the function SCTransform() was used in place of standard preprocessing methods[19].

The FindAllMarkers() function was used on each dataset to generate markers for each cluster. Cell types and nuclear identities were assigned based on a combination of previously reported marker genes along with gene ontology analysis of uniquely expressed genes. Gene enrichment or differential expression was considered significant with an adjusted $p$ value < 0.05. Regarding gene ontology, the top 100 marker genes of each cluster were analyzed using ToppGene[56].

In order to integrate multiple snRNA-seq datasets, the preprocessed datasets to be integrated were run through the FindIntegrationAnchors() function. Resulting integration anchors were used in the IntegrateData() function, creating a new integrated Seurat object. Following this, the standard workflow for visualization and clustering was used. To generate UMAPs of cell identities split or grouped based on their original dataset, the parameters split.by and group.by were set to "orig.ident". Heatmaps were generated from subpopulations using the DoHeatmap() function. For the heatmap comparing gene expression across aging (5 months/24 months/30 months), a custom heatmap was created and the full list of differentially expressed genes was reported in Supplemental Data 3. In order to determine differential expression between populations of interest across datasets, the function FindMarkers() was used. In order to identify genes that were consistently changed in aging myonuclei, pairwise comparisons of each aging dataset with the 5-month dataset were performed, and a common aging gene signature was identified. To assess integration of the data, without batch correction, we performed an additional unsupervised analysis using the software ICGS2 (http://altanalyze.org). ICGS2 was run using the default parameters, with cell cycle removal on all combined timepoint snRNA-Seq samples.

To ensure the absence of artifactual clusters from nuclei doublets, we computationally predicted heterotypic doublets using the software DoubletDecon[57]. We examined marker genes (MarkerFinder algorithm) within each Seurat3 produced set of clusters and subclusters and combined main clusters. By definition, no large clusters can be doublets (>20% of the size of the largest nuclei population). No small-to-medium sized primary and subclusters were enriched in putative DoubletDecon predictions (>70%) or lacked unique marker genes, suggesting such populations were not doublets.

To identify transcription factor (TF) binding sites that are enriched within the promoter regions of $Nos1^+$ and $Enah^+$ genes, we used the HOMER motif enrichment algorithm[58] and a large library of human position weight matrix (PWM) binding site models obtained from the CisBP database, build 2.0[59]. We used a definition of 1000 bp upstream and downstream of the transcription start site (TSS) to generate consensus predictions.

**Single-molecule RNA fluorescent in situ hybridization**. Single-molecule FISH experiments were performed using RNAscope (ACDBio) following the manufacturer's protocols. We used fresh-frozen cross-sections of quadriceps or gastrocnemius muscles where indicated, as well as fresh-frozen longitudinal sections of tibialis anterior. Following the completion of the RNAscope fluorescent assay, an immunostaining step was carried out to label myofibers. Sections were blocked for 30 min using 1% BSA, 1% heat-inactivated goat serum, and 0.025% Tween-20/PBS,

followed by incubation for 30 min at RT with anti-laminin (1:100; Sigma L9393). Secondary AlexaFluor antibodies (1:200) (Invitrogen) were then applied at room temperature for 1 h. For visualization of neuromuscular junctions, an α-Bungarotoxin AlexaFluor conjugate (1:100; Thermo Fisher) was added with the secondary antibody. For staining of muscle fibers, we used anti-myosin primary antibodies (MF 20, 1:20, Developmental Studies Hybridoma Bank) and incubated for 1 h at RT. Sections were mounted using VectaShield with DAPI (Vector Laboratories). Slides were imaged using a Nikon A1R confocal system.

**Cell culture**. C2C12 cells were acquired from American Type Culture Collection. Cells were grown in DMEM (Gibco) containing 10% heat-inactivated bovine growth serum (BGS) and supplemented with penicillin-streptomycin (1%), and incubated at 37 °C with 5% $CO_2$. C2C12 cells were differentiated by switching medium to DMEM with 2% heat-inactivated horse serum with antibiotics. In order to induce formation of aneural AChR clusters, 35 mm plates were precoated with 10 µg/ml natural mouse laminin protein (ThermoFisher 23017015) suspended in L-15 medium (Thermo Fisher) with 0.2% $NaHCO_3$[50]. Cells were transfected with siRNAs on day 2 of differentiation using Lipofectamine 2000 (Thermo Fisher) according to the manufacturer's protocol. All siRNAs were purchased from Santa Cruz.

On day 5 of differentiation, cells were rinsed with PBS and processed for RNA isolation or immunocytochemistry. For the latter, cells were fixed in 4% paraformaldehyde for 20 min at room temperature. Cells were then permeabilized with 0.2% Triton X-100/PBS for 20 min at room temperature, followed by blocking in 3% BSA/PBS for 30 min. We then incubated with phalloidin (1:100, Thermo Fisher) and α-Bungarotoxin (1:200, Thermo Fisher) AlexaFluor conjugates for 1 h at room temperature. Cells were then rinsed with PBS and nuclei were stained with Hoechst 33342 solution (Thermo Fisher). Imaging was performed using an upright Nikon FN 1 microscope on a Nikon A1R confocal.

**Quantitative real-time PCR**. RNA was isolated from homogenized muscle tissue and plated cells using Trizol reagent (Invitrogen) according to standard protocols. cDNA synthesis was carried out using the MultiScribe™ kit (Applied Biosystems) and qPCR reactions were performed on a Bio-Rad CFX96™ Real-Time System using PowerUp™ SYBR Green Master Mix (Applied Biosystems). Primers used for qPCR are listed in Supplementary Table 1.

**Statistics**. GraphPad Prism 8 software was used for statistical analysis, excluding single-nucleus RNA-sequencing data. Unpaired two-sided t-tests or one-way ANOVAs with Tukey post-hoc comparison were used to determine statistical significance, which was set at p values < 0.05. For qPCR analysis, ΔCt values were used to assess significance, and ΔΔCt values were graphed to show relative expression compared to control.

**Reporting summary**. Further information on research design is available in the Nature Research Reporting Summary linked to this article.

## Data availability

Raw sequencing data are deposited in GEO (GSE147127). Additional data files and metrics have been deposited at https://www.synapse.org/#!Synapse:syn21676145. Source data are provided with this paper. An interactive data portal is publicly available at https://research.cchmc.org/myoatlas.

## Code availability

Data analysis was limited to standard pipelines in both Seurat and SoupX R packages, as described in the Methods. Code is available at https://github.com/MillayLab/single-myonucleus

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

## Acknowledgements
We thank the following entities at Cincinnati Children's Hospital Medical Center: Kelly Rangel and Shawn Smith from the Gene Expression Core, David Fletcher at the Sequencing Core, Steven Potter, and Hee-Woong Lim. The main funding source for this work was from the Research Innovation and Pilot Funding Program of the Cincinnati Children's Hospital Research Foundation to D.P.M. and N.S. This work was also supported by grants to D.P.M. from the National Institutes of Health (R01AR068286, R01AG059605) and Pew Charitable Trusts. A Cincinnati Children's Hospital Endowed Scholar Award supported D.P.M. and M.T.W. M.T.W. laboratory was also supported by grants from the National Institutes of Health (R01NS099068, R01AR073228, R01GM055479). C.O.S. was supported by an undergraduate fellowship from the American Heart Association (18UFEL33930019). M.J.P. was supported by a training grant from the National Institutes of Health (NHLBI T32HL007752).

## Author contributions
M.J.P. and D.P.M. designed experiments. M.J.P., C.O.S., C.Su., X.C., and K.C. performed experiments. M.T.W. and N.S. provided bioinformatic guidance. All authors analyzed the data and contributed to interpretations. M.J.P. and D.P.M. wrote the manuscript with assistance from all authors.

## Competing interests
The authors declare no competing interests.
