## [Peer Review File · Nature Communications]

Reviewers' Comments:

Reviewer #1:

Remarks to the Author:

In the revision of the manuscript, Michael J Petrany et al. replied to some points suggested by the reviewer. The populations and sub-populations of nuclei that they identified is undoubtedly interesting in the muscle field, however the data in support of the new clusters are still poor. Especially for the clusters that are labelled with the name of their marker genes, their functions in the muscle are only superficially suggested. More advanced bioinformatic analyses could help to better define the lineage and the peculiarities of these nuclei, but, as suggested by authors, analyses such as pseudo-time analysis or RNA velocity do not fit with the generated data. To reach the standard level of the journal, the biological roles of these clusters of nuclei has to be verified and explored more in depth. Otherwise the paper remains descriptive with limited mechanistic insight.

Reviewer #2:

Remarks to the Author:

The authors have addressed adequately this Reviewer's concerns. The manuscript has been improved and it is very solid. It will be an important contribution to the skeletal muscle field.

Reviewer #3:

Remarks to the Author:

The authors have addressed all my concerns. Thank you.

Reviewer #4:

Remarks to the Author:

I'm overall satisfied with most of the point by point response. However, I'm still having concerns about the clustering results in Fig S5a-b. In the method part, it's not clear how the authors decided the proper clustering resolution in the Seurat pipeline to avoid potential overclustering, nor did they state why Flnc and Enah, rather than those top differentially expressed gene listed in the heatmap of Fig S5, were chosen for the validation of subtype diversity within myonuclei. Also, despite showing the interesting temporal changes of Flnc and Enah expression in Fig 3b, no statistic test has been done to determine the differences of their expression levels among those subclusters (Fig S5c).

Additionally, instead of just listing marker genes, could the authors also provide more biological interpretation of the data with analysis such as GO, GSEA or motif enrichment to demonstrate the differences in structure/function/regulation between different types of myofibers?

Reviewer #1 (Remarks to the Author):

In the revision of the manuscript, Michael J Petrany et al. replied to some points suggested by the reviewer. The populations and sub-populations of nuclei that they identified is undoubtedly interesting in the muscle field, however the data in support of the new clusters are still poor. Especially for the clusters that are labelled with the name of their marker genes, their functions in the muscle are only superficially suggested. More advanced bioinformatic analyses could help to better define the lineage and the peculiarities of these nuclei, but, as suggested by authors, analyses such as pseudo-time analysis or RNA velocity do not fit with the generated data. To reach the standard level of the journal, the biological roles of these clusters of nuclei has to be verified and explored more in depth. Otherwise the paper remains descriptive with limited mechanistic insight.

Response: We agree with this reviewer that the precise functions of the identified nuclei clusters are not fully elucidated. Determining the biological roles of these clusters will require an entire new series of reagents, likely genetically modified mice, which will be the focus of future work. We added this caveat to the final paragraph of the discussion - "One caveat of our work is the lack functional validation of myonuclear populations, which will be the focus of future investigations."

Reviewer #2 (Remarks to the Author):

The authors have addressed adequately this Reviewer's concerns. The manuscript has been improved and it is very solid. It will be an important contribution to the skeletal muscle field.

Response: We are pleased to hear the work is viewed as 'very solid' and 'an important contribution to the skeletal muscle field'.

Reviewer #3 (Remarks to the Author):

The authors have addressed all my concerns. Thank you.

Response: Thank you.

Reviewer #4 (Remarks to the Author):

I'm overall satisfied with most of the point by point response. However, I'm still having concerns about the clustering results in Fig S5a-b. In the method part, it's not clear how the authors decided the proper clustering resolution in the Seurat pipeline to avoid potential overclustering, nor did they state why Flnc and Enah, rather than those top differentially expressed gene listed in the heatmap of Fig S5, were chosen for the validation of subtype diversity within myonuclei. Also, despite showing the interesting temporal changes of Flnc and Enah expression in Fig 3b, no statistic test has been done to determine the differences of their expression levels among those subclusters (Fig S5c).

Response: We agree with the reviewer that these clusters are somewhat speculative. Our goal of this analysis was to test if a deeper interrogation of myonuclear populations at 5 months of age, which is a time point when the muscle is fully grown and is mainly in general homeostasis, would reveal heterogeneity. We provide multiple clarifications below, in order to make this analysis clearer and we also significantly altered the figure and text to attain clarity for the reviewer and readers.

1. In terms of resolution in Seurat, for all of our analysis we used a resolution of 0.5 to generate clusters, which is a conservative standard resolution in the pipeline, and we have added that in the Methods. To test if clusters were maintained with a lower resolution, we reduced it to 0.2 and here we found that 4 clusters were merged into 2. We decided to keep the clusters generated from the 0.5 resolution because that is consistent with the rest of our analysis. We do, however, significantly adjust our interpretation of these results and comment on these resolution differences (see point 3).
2. We also have applied two orthogonal independent bioinformatic approaches (ICGS2 and cellHarmony) to evaluate the reliability of these clusters. Unique transcriptionally distinct populations of cells, in principle, should be defined both by uniquely present transcripts with increased relative abundance in every cell population and should be sufficiently stable to enable re-identification through an orthogonal unsupervised and supervised approach. Stability can be defined by using supervised classification of the centroids/markers of the clusters (to determine the fidelity at which the same cells are assigned back to the original clusters) or through independent and robust unsupervised approaches. We applied each of these evaluations to our sub-clusters of Type IIb and Type IIx myonuclei.

From these analyses, we observe medium-to-strong evidence supporting the validity of these clusters from: 1) the detection of unique marker genes, 2) supervised iterative re-classification and 3) orthogonal unsupervised analyses. For Type IIb myonuclei, unique marker genes were identified for all cell clusters using the recommended minimum MarkerFinder marker specificity cutoff of $\rho > 0.2$ (AltAnalyze). When re-classifying the cells in the dataset against its own centroids using the software cellHarmony, 3 out of the 4 clusters had a percent correct assignment of $\geq 75\%$ (c1 = 48% correct). When compared to an unsupervised analysis in the software ICGS2, all four clusters had top-10 marker genes that corresponded to the top marker genes from ICGS2 (n=6 clusters).

For the Type IIx myonuclei, we observed strong evidence to support the stability of 3 out of the four clusters using the three criterion described above, with weaker evidence for a fourth cluster (c1). All clusters had unique MarkerFinder markers using the above described criteria. Two of the four clusters had cellHarmony percent correct assignments for $\geq 75\%$ of cells (c2 and c3), while c0 and c1 were correct for 49% and 43%. ICGS2 identified analogous clusters for

3 out of 4 of the Seurat clusters (c0, c2 and c3) based on the top expressing markers, suggesting that c1 may be a result of over-clustering.

These analyses suggested confirmation of the validity of four distinct Type IIb myonuclei clusters supported by unique marker gene expression, independent unsupervised clustering and supervised re-classification. While all four of the observed Type IIx myonuclei subclusters were evidenced by unique marker gene expression, cluster 1 lacked strong evidence for being called a separate cluster (no orthogonal unsupervised cluster or sufficient supervised re-classification evidence). Because this could be classified as medium evidence, and we are unable to provide more direct confirmation, we decided to adjust our interpretation of the data (see point 3).

3. Based on points 1 and 2, we made certain to not over-interpret these results and in the revised version we alter the tone of the paragraph (third paragraph of Results section) discussing this data. Specifically, we suggest there could be modest diversity, which motivated us to investigate other muscles and time points, but do not state that the data indicate heterogeneity. Here is the explanation in the Results section: "Sub-clustering revealed multiple nuclear compartments with divergent transcriptional states (Supplementary Fig. 5b, c). While these clusters do not possess many unique markers, they do exhibit a spectrum of gene expression potentially indicating modest transcriptional heterogeneity in homeostatic muscle. We then investigated the possibility of stronger myonuclear sub-types with distinct functions utilizing different types of skeletal muscle and at varying ages."
4. We decided to remove the *Flnc* and *Enah* violin plots because we agree with the reviewer that it is not intuitive as to why those were chosen over other potential markers. The initial reason for choosing *Flnc* and *Enah* was based on their presence in our P21 and aged data. Once this sarcomeric signature was identified at P21 we asked if there was a small hint of that signature in fully developed 5 month old muscle. Since this was a retrospective decision and we agree that the presence of *Flnc* and *Enah* at 5 months, by itself, is not strong evidence for transcriptional heterogeneity, we decided to remove the data.

Additionally, instead of just listing marker genes, could the authors also provide more biological interpretation of the data with analysis such as GO, GSEA or motif enrichment to demonstrate the differences in structure/function/regulation between different types of myofibers?

Response: As requested, we performed gene set enrichment analyses on the various myofiber types that comprise the TA muscle at 5 months of age, specifically Type IIx (*Myh1⁺*) and Type IIb (*Myh4⁺*). Here we used the software ToppGene (<https://toppgene.cchmc.org>) and found GO process and cellular localization terms associated with sarcomeres and muscle contraction that are similar between the two

sets of myofibers. The major difference was associated with oxidative metabolic properties of Type IIx myofibers. Indeed, it is known that Type IIb (*Myh4*⁺) myofibers are fast contracting and glycolytic, whereas Type IIx (*Myh1*⁺) myofibers possess more characteristics of oxidative metabolism. This validates the nuclear populations and the GO is shown in Fig. S5a. We now write: “Gene Ontology enrichment analysis confirmed that these nuclear populations are enriched in cellular component terms associated with the sarcomere and muscle contraction, while terms associated with mitochondria were selectively present in *Myh1*⁺ nuclei, which are known to have an increased oxidative metabolic profile (Supplementary Fig. 5a). Sub-clustering revealed multiple nuclear compartments with divergent transcriptional states (Supplementary Fig. 5b, c).”

Reviewers' Comments:

Reviewer #4:

Remarks to the Author:

The authors have satisfied my comments.

Reviewer #4 (Remarks to the Author):

The authors have satisfied my comments.

Author response: Thanks!